# Photochemistry and Photophysics of Cholesta-5,7,9(11)-trien-3β-ol in Ethanol

**DOI:** 10.3390/molecules28104086

**Published:** 2023-05-14

**Authors:** Jack Saltiel, Sumesh B. Krishnan, Shipra Gupta, Anjan Chakraborty, Edwin F. Hilinski, Xinsong Lin

**Affiliations:** Department of Chemistry and Biochemistry, Florida State University, Tallahassee, FL 32306-4390, USA; skrishnan2@fsu.edu (S.B.K.); shigupta@valdosta.edu (S.G.); anjanc@iiti.ac.in (A.C.); hilinski@chem.fsu.edu (E.F.H.); xlin@chem.fsu.edu (X.L.)

**Keywords:** cholestatrienol, photochemistry, photophysics, fluorescence probe

## Abstract

Cholesta-5,7,9(11)-trien-3β-ol (9,11-dehydroprovitamin D_3_, CTL) is used as a fluorescent probe to track the presence and migration of cholesterol in vivo. We recently described the photochemistry and photophysics of CTL in degassed and air-saturated tetrahydrofuran (THF) solution, an aprotic solvent. The zwitterionic nature of the singlet excited state, ^1^CTL* is revealed in ethanol, a protic solvent. In ethanol, the products observed in THF are accompanied by ether photoadducts and by photoreduction of the triene moiety to four dienes, including provitamin D_3_. The major diene retains the conjugated *s*-*trans*-diene chromophore and the minor is unconjugated, involving 1,4-addition of H at the 7 and 11 positions. In the presence of air, peroxide formation is a major reaction channel as in THF. X-ray crystallography confirmed the identification of two of the new diene products as well as of a peroxide rearrangement product.

## 1. Introduction

Cholesta-5,7,9(11)-trien-3β-ol (CTL) and its close relative, 9(11)-dehydroergosterol (DHE) are natural products [1,2,3,4,5,6] that possess fluorescent [7] conjugated triene moieties embedded in the rigid cholestane skeleton.

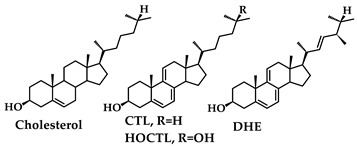


CTL differs from cholesterol only by the two additional double bonds at the C_7_ and C_9_ positions and DHE also differs in the side chain at C_17_. Consequently, CTL and DHE have been used as fluorescence probes for studies of cholesterol trafficking and membrane organization in vivo [8,9,10,11]. Because CTL and DHE are photochemically active, their fluorescence intensity decreases on prolonged excitation leading to photobleaching and fluorescence recovery studies [12]. Our initial studies of the photochemistry and photophysics of CTL and its 25-hydroxy derivative, HOCTL, were carried out in tetrahydrofuran (THF), an aprotic solvent [13]. The presence of the OH substituent at C_25_ of CTL led to facile crystallization of photoproducts and allowed us to confirm results from early pioneering studies by the Windaus and Barton groups [14,15,16] using X-ray crystallography (Figure 1) [13]. The peroxide had been obtained from DHE by eosin sensitization [14,15] and its structure was proposed by Fieser [17] before the role of ^1^Δ_g_ singlet oxygen (^1^O_2_*) in such reactions was known. ^1^O_2_* luminescence observed on irradiation of an air-saturated solution of CTL in CCl_4_ established the formation of CTL triplets that transfer energy to ^3^O_2_ [18].

Followed by UV, the loss of CTL or HOCTL in THF is accompanied by the appearance of a structureless peak at 261 nm due to the formation of the rearranged products RP_1_ or HORP_1_ [13]. In contrast, a 1930 study found that upon direct excitation of DHE in ethanol, the *λ*_max_ of the photoproduct mixture is at 243 nm [19]. We resolve this discrepancy by showing that on irradiation of HOCTL in degassed ethanol the rearranged product (HOP_1_, 37%) is accompanied by formation of at least two ethers: 6α-ethoxy-3β-cholesta-7,9(11)-diene-3,25-diol (HOE_1_, 24.5%) and 6β-ethoxy-3β-cholesta-7,9(11)-diene-3,25-diol (HOE_2_, 13.4%); and four dienes: 3β-cholesta-7,9(11)-diene-3,25-diol (HOD_1_, 9.1%), 3β-cholesta-5,7-diene-3,25-diol (HOPro, 3.0%), 3β-cholesta-5,8-diene-3,25-diol, (HOD_2_ 0.7%), and 3β-cholesta-6,8-diene-3,25-diol (HOD_3_, 8.2%) (Figure 2). In the presence of air, at high CTL concentrations, peroxide formation is the dominant reaction in both solvents. The 25-OH substituent does not influence the photochemistry or photophysics. During the HPLC purification of HOCTLP, we isolated a hitherto unknown rearrangement product.

## 2. Results

### 2.1. Photochemical Observations

A degassed 3.0 mL aliquot of HOCTL (5.43 × 10^−5^ M, contaminated with 2% HOPro) in ethanol was irradiated in a 13 mm Pyrex test tube provided with a sidearm to which a 0.50 × 1.0 cm quartz cell was attached via a graded seal. Figure 1 shows the UV evolution of HOCTL irradiated at λ_exc_ = 313 nm at 24.0 °C (time sequence 0, 10, 20, 30, 40, 60, 80, 100, 120, 180, 270 and 360 min). The photoproduct mixture has λ_max_ = 244 nm and an isosbestic point is maintained at 279 nm. Similar results were obtained in parallel experiments on CTL.

### 2.2. Photoproducts

For product isolation and identification, we irradiated 520 mL of a HOCTL (2.65 × 10^−4^ M, contaminated with 5% HOPro) solution in a Hanovia reactor using a 450 W Hanovia Hg lamp and the 313 nm filter solution. The solution was outgassed with Ar for 1.0 h prior to the irradiation and a constant flow of Ar was maintained during the irradiation. The progress of the photoreaction was monitored by UV-vis spectroscopy and near completion was achieved after 6 h. The solution was concentrated with a Büchi rotary evaporator and taken to dryness to remove ethanol using a vacuum pump. The ^1^H NMR of the crude mixture was recorded in CD_3_OD, (Appendix A). The mixture was then dissolved in 1.0 mL CHCl_3_ and injected onto the semi-preparative HPLC column. Initial elution with hexane/ethanol with ethanol gradually increasing from 0 to 3% was followed by up to 70% ethanol to wash all ethers from the column. The flow rate was 5 mL/min, 5.0 mL fractions were collected and the progress of the separation was followed by UV. Separation of HOPro, HOCTL, HORP_1_, HOD_1_ and HOD_2_ was challenging due to very similar retention times. HOPro, HOD_1_, HOCTL, and HORP_1_ eluted in fractions containing 1.43–1.45% ethanol. The earliest fractions containing mainly HOPro were discarded. Fractions rich in HOD_1_ were in the 1.43–1.44% ethanol range and fractions rich in HORP_1_ eluted with 1.44–1.45% ethanol. HOD_1_ fractions were combined, concentrated, dried and dissolved in 1.0 mL CHCl_3_ and separately chromatographed using hexane/ethyl acetate as the solvent system with ethyl acetate gradually increasing from 0 to 20%. Pure HOD_1_ was collected in 2–3 mL fractions at 14.8% ethyl acetate. HORP_1_ fractions were also combined and subjected to a third chromatography with hexane/ethyl acetate eluent. Pure HORP_1_ was collected in 0.6 mL fractions at 11.5% ethyl acetate. The final fractions from the first chromatography containing HOE_1_ and HOE_2_ were combined, concentrated, dried, dissolved in 1 mL CHCl_3_ and re-chromatographed using the hexane/ethanol solvent system. Pure HOE_1_ and pure HOE_2_ were collected in fractions containing 2.4 and 2.6% ethanol, respectively. The ^1^H NMR of pure HORP_1_ above was identical to the spectrum we reported for the rearrangement product in THF [13]. Its UV spectrum was also identical to the spectrum we observed in THF except that its *λ*_max_ shifted from 261 nm in THF to 259 nm in ethanol. The UV spectra of the conjugated *s*-*trans*-diene, HOD_1_, *λ*_max_ = 242 nm, and of the *s*-*trans*-diene ether products, HOE_1_ and HOE_2_, *λ*_max_ = 243 nm, account for the *λ*_max_ of the product mixture in Figure 1. Windaus et al. reported *λ*_max_ = 245 nm [20] for 3*β*-Δ^7,9^-cholestadienol and a more recent measurement gave *λ*_max_ = 243 nm [21,22]. The UV spectra of HOD_1_ and HOE_1_ are very similar (Figure 2), reflecting the shared presence of the *s*-*trans*-diene chromophore. The UV spectrum of HOE_2_ is relatively structureless (Appendix A). Isolation of the unconjugated diene, HOD_2_, was serendipitous as it appeared as crystals in one of the chromatography fractions.

^1^H NMR spectra of the photoproducts are shown in Appendix A. They are as follows: Appendix A, **HOD_1_** (CD_3_OD, 500 MHz, *δ*): 5.49 (1H, d), 5.39 (1H, s), 3.50 (1H, m), 2.27–2.35 (1H, dd), 2.18 (1H, t), 2.11 (1H, d), 1.98 (2H, m), 1.89 (2H, m), 1.74–1.87 (2H, m), 1.69 (1H, m), 1.30–1.53 (17H, m), 1.17 (6H, s), 1.07 (1H, m), 0.97 (3H, d), 0.92 (3H, s), 0.54 (3H, s). Appendix A, **HOD_2_** (CDCl_3,_ 500 MHz, *δ*): 5.43 (1H, t), 3.54 (1H, m), 2.53 (2H, m), 2.26–2.38 (2H, m), 2.06–2.21 (3H, m), 1.96–2.03 (1H, m), 1.83–1.95 (4H, m), 1.26–1.70 (23H, m), 1.21 (3H, s), 1.19 (3H, s), 0.99–1.10 (3H, m), 0.95 (3H, s), 0.65 (3H, s). Appendix A, **HOE_1_** (CD_3_OD, 500 MHz, *δ*): 5.57 (1H, d), 5.43 (1H, s), 3.73 (1H, m), 3.58 (1H, d), 3.47 (2H, m), 2.35 (1H, dd), 2.10–2.27 (4H, m), 1.9–2.06 (3H, m), 1.77–1.88 (3H, m), 1.23–1.55 (18H, m), 1.18–1.23 (3H, t), 1.17 (8H, s), 0.97 (6H, t) 0.55 (3H, s). Appendix A, **HOE_2_** (CD_3_OD, 500 MHz, *δ*): 5.67 (1H, d), 5.40 (1H, s), 3.91 (1H, t), 3.67 (1H, s), 3.60 (2H, m), 2.40 (1H, dd), 2.36–2.43 (1H, dd), 2.22–2.28 (1H, t), 2.16–2.22 (1H, d), 1.94–2.06 (3H, m), 1.59–1.85 (5H, m), 1.23–1.58 (18H, m), 1.15–1.23 (13H, m), 1.07 (m, 2H), 0.97 (3H, m), 0.91 (1H, m), 0.61 (1H, s).

^1^H NMR spectra of HOD_1_ and HOD_2_, but without the 25-OH group, are known [23]. The *α*-ethoxy assignment to HOE_1_ is based on its COSY and NOESY ^1^H NMR spectra. The COSY spectrum (Appendix A) allowed the assignment of the C_6_ proton at *δ* 3.58 (d, J 10 Hz–due to coupling to the C_5_ proton) based on its being correlated to the C_7_ vinyl proton at *δ* 5.43 (broad s) (Appendix A (expanded region of Appendix A)). The vinyl H at C_11_ at *δ* 5.57 (d) is correlated with the C_12_ protons at *δ* 2.35 (dd) (Appendix A (expanded region of Appendix A)). The proximity of the C_6_ proton to the protons of the C_17_ methyl group, revealed by the NOESY spectrum (Appendix A) established its *β* orientation and the *α* disposition of the ethoxy group. Additional support for our assignment is provided by the ^1^H NMR spectra in CDCl_3_ of the analogous Δ^7^ compounds shown below [24].

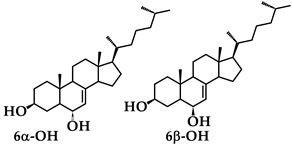


The vinyl Hs at C_7_ of 6*α*-OH and 6*β*-OH appear at *δ* 5.18 (br s) and 5.45 (br d), respectively, and the Hs at C_6_ appear at 3.8 (m) and 3.92 (m) in the same order. We observed the same chemical shift difference in the same direction for the C_6_ proton in HOE_1_ (*δ* 3.58) and HOE_2_ (*δ* 3.68), Also significant is the appearance of the C_7_ vinyl Hs of 6*α*-OH and HOE_1_ as broad singlets and the C_7_ vinyl Hs of 6*β*-OH and HOE_2_ as broad doublets.

HOCTLP, a major product from the irradiation of HOCTL in ethanol in the presence of air [13], was also purified by semi-preparative HPLC. During a purification run, we noticed the formation of small crystals in one of the fractions. Recrystallization from ethanol gave X-ray quality needle-like crystals that correspond to a peroxide rearrangement product, RP_2_, with a 5-membered A ring and a 7-membered B ring (Figure 3). The ^1^H NMR spectrum of the compound recorded in CDCl_3_ is shown in Appendix A. Characteristic peaks in the ^1^H NMR spectrum of RP_2_ in CDCl_3_ were a singlet at *δ* 13.94 for the hydrogen-bonded OH proton and a triplet at *δ* 5.460 for the single vinyl proton.

### 2.3. X-ray Structures

We concentrated our studies on HOCTL because of the relative ease of obtaining good-quality crystals when the 25-hydroxy group is present. Definitive structure assignment to the two diene photoproducts, HOD_1_ and HOD_2_, was achieved using X-ray crystallography (Figure 3). The X-ray crystal structure of the peroxide rearrangement product is shown in Figure 4. Mercury cif files and structural details are provided in Appendix A.

### 2.4. Photoproduct Quantum Yields

The quantum yield measurement of HORP_1_ formation in ethanol was carried out in parallel with the previously described experiment [13] in THF. HOCTL contaminated with 11% Pro based on ^1^H NMR and UV analysis was used. Three-milliliter aliquots of degassed HOCTL solutions, 4.01 × 10^−3^ M in ethanol, were irradiated in the Moses merry-go-round [25] in parallel with the 3.0 mL 4.27 × 10^−2^ M degassed pentane solution of *trans*-stilbene, t-St, (0.011% *cis*-stilbene by GC), used for actinometry. The samples were immersed in a water bath whose temperature was maintained at 25 °C. A 450 W Hanovia lamp was employed together with the 313 nm filter solution. Pro is transparent at *λ*_exc_ = 313 nm and does not interfere with these measurements. Following 12 h irradiation, HOCTL conversions to HORP_1_ based on the relative areas of the vinyl protons being 7.4% in ethanol (^1^H NMR), while *trans*- → *cis*-stilbene conversion corrected for backreaction [26] was 12.38 *±* 0.23% *cis*-stilbene. With the use of *ϕ*_tc_ = 0.52 in the actinometer [25], we obtain *ϕ*_HORP1_ = 0.024 in ethanol. This experiment was repeated with pure HOCTL, 1.0 × 10^−3^ M, and [*t*-St] = 4.01 × 10^−2^ M, as above, except that the irradiation was carried out with a 550 W lamp at 26.4 °C. Conversions were 8.33% HORP_1_ and 4.46% c-St (corrected) confirming the *ϕ*_HORP1_ = 0.024 value.

The presence of an isosbestic point in the UV spectra in Figure 1 and the fact that spectral matrices obtained in similar experiments on PCA-SM treatment behave as robust two-component systems (see Appendix A), shows that the product composition is maintained throughout the photoreaction. Therefore, the final photoproduct composition reflects the relative formation quantum yields. The experiment was repeated without any HOPro contaminant in order to establish HOPro as a photoproduct. The vinyl and the C_18_ methyl protons contribute almost uniquely to the spectra and were used to determine the product composition. Expanded NMR regions in CD_3_OD from the irradiation of pure HOCTL are shown in Figure 5. Quantum yields were obtained from product contributions, determined by cutting and weighing in triplicate the vinyl and C_18_ methyl peaks, and setting the masses of the HORP_1_ peaks equal to a quantum yield of 0.024. Only in the case of HOD_2_ were chemical shifts based on spectra measured in CDCl_3_ (see Appendix A). The two ^1^H NMR analyses give remarkably similar product yields (see the Appendix A), the averages of which are shown in Table 1. The isolated photoproducts account for 88% of HOCTL loss.

In the presence of air, peroxide formation becomes the dominant photoreaction. Furthermore, as in THF [13], the relative yield of peroxide increases at higher CTL or HOCTL concentrations. Measurements in ethanol were carried out in parallel with those described for THF [13]. Peroxide quantum yields as a function of [HOCTL] are given in Table 2.

### 2.5. Fluorescence Measurements

We determined the effect of O_2_ on HOCTL fluorescence intensity and fluorescence lifetime by measuring the fluorescence of Ar-bubbled, air-saturated and O_2_-bubbled ethanol solutions (Table 3). For the fluorescence quantum yield measurements, the temperature in the jacketed cuvette was maintained at 20.0 °C by circulating cold water from a Neslab RTE-4DD heating/cooling bath. Values measured at 25 °C, are also shown in Table 3. For the lifetime measurements, samples were excited with a 295 nm, <0.75 ns diode pulse, and fluorescence decay was monitored at 367 nm at room temperature T (20.3 °C). Average lifetime values and the range of *χ*^2^ values are given in Table 3. A second minor (3–5%) longer-lived component (3–6 ns) was ignored, as it is probably due to an impurity. We calculated the [O_2_] values in Table 3 for air- and O_2_-saturated ethanol using O_2_ mole fraction data [27], the density [28] and the vapor pressure of ethanol at 20 °C [29].

### 2.6. HOCTL Triplet

A 3 mL aliquot of an ethanol solution containing 3.79 × 10^−3^ M HOCTL and 2.5 × 10^−3^ M benzophenone was transferred into a Pyrex tube equipped with a standard taper joint, provided with a grease trap and attached via a graded seal to a standard 1.0 cm^2^ UV cell. The solution was degassed using six freeze-pump-thaw cycles to <10^−4^ Torr and flame-sealed at a constriction. Pulse excitation at 370 nm led to transient spectra recorded periodically in the 0–100 μs time scale. As in THF [13], in addition to the known transient absorptions of the benzophenone triplet and the benzophenone ketyl radical [30,31], we observed a new transient absorption, assigned to ^3^HOCTL*, with *λ*_max_ = 390 nm and a shoulder at 408 nm (Figure 6a) slightly blue-shifted from the values in THF [13]. Decay monitored at 395 nm, Figure 6b, is monoexponential and gives *τ* = 73.5 μs.

## 3. Materials and Methods

### 3.1. Materials

Sources and purifications of 7-dehydrocholesterol (provitamin D_3_, Pro), 25-hydroxyprovitamin D_3_ (HOPro) as well as the syntheses and purifications of their 9(11) dehydro derivatives, CTL and HOCTL, were previously described [13]. Ethanol (Koptec, 200 proof, absolute/anhydrous), hexane (EMD, HPLC grade), CHCl_3_ (EMD, HPLC grade) and ethyl acetate (EMD, HPLC grade) were used as received.

### 3.2. Analytical Methods

Analysis of product composition was by GC, ^1^H NMR, UV-vis, fluorescence, MS and GC/MS, as previously described [13]. We employed a Beckman Ultrasphere 5 μm Si 250 × 10 mm column for semi-preparative HPLC purifications. The solvent systems used were hexane/ethyl acetate and hexane/ethanol for HOCTL photoproducts: 5 mL/min, monitored at 240 or 280 nm. UV-vis absorption spectra were measured on a Varian Cary 300B spectrometer. Fluorescence spectra were measured using a Horiba Fluoromax 4 fluorometer. Fluorescence lifetimes were determined using a different Horiba Fluoromax 4 instrument equipped with a time-correlated single-photon counting accessory and an R928 PMT detector (Hamamatsu). The light source was a 296 nm nanoLED (Horiba) having a pulse duration of <0.75 ns and a 1 MHz repetition rate. The photon count was set at 10,000 and the time-to-amplitude converter range was set at 50 ns. The instrument response function was obtained by collecting Rayleigh scatter at 296 nm. Fluorescence lifetimes were determined by reconvoluting the instrument response function with exponential decay using DAS6 (Horiba) fluorescence decay analysis software. The quality of the fits was judged by *χ*^2^ values, standard deviations of derived lifetimes and visual inspection of the residuals. Absolute fluorescence quantum yields were measured with the use of a Hamamatsu Quantaurus-QY spectrometer equipped with a 150 W Xenon arc lamp. The instrument employs an integrated sphere sample chamber with a cooled back-thinned 1024-channel charge-coupled device sensor as the detector. This avoids the need for a fluorescence standard by using the attenuation in the area of the Rayleigh scattered light peak at *λ*_exc_ = 325 nm to measure photon absorption. HOCTL solutions were outgassed with oxygen or argon for 2 h prior to quantum yield and lifetime measurements. Transient absorption measurements and triplet decay kinetics were measured using an Edinburgh Instruments LP980-KS Laser Flash Photolysis spectrometer. A Continuum Nd:YAG laser provided the excitation pulse and transient absorption was monitored with a 150 W Xenon lamp. A Bruker 500 MHz NMR spectrometer was used to measure NMR spectra in CDCl_3_ and CD_3_OD.

### 3.3. Crystallography

HOD_1_ and HOD_3_ crystals, approximately 0.33 × 0.24 × 0.07 mm^3^, were adhered to a MiTeGen loop with Paratone oil. Crystallographic data were collected at 150 K on a Rigaku-Oxford Diffraction XtaLAB-Synergy-S diffractometer with a Hypix-6000HE (Hybrid Photon Counting) detector, using Cu-Kα radiation of wavelength 1.54187 Å. The intensity data were measured by ω-scan with 0.5° oscillations for each frame with an intensity greater than 10:1 for the data-to-parameter ratio. The program suite CrysAlis^Pro^ was used for data collection, absorption correction, and data reduction. The structures were solved with the dual-space algorithm using SHELXT and were refined by full-matrix least-squares methods on *F*^2^ with SHELXL-2014 using the GUI Olex2 program [32]. HOP_2_ and HOP_3_, C_27_H_44_O_2_, crystallized in the orthorhombic crystal system, with the space group *P*2_1_2_1_2_1_. All non-hydrogen atoms were refined anisotropically. Hydrogen atoms were inserted at calculated positions or, if possible, based on difference Fourier analysis, and refined with a riding model or without restrictions.

### 3.4. Irradiation Procedures

Sample preparation and degassing procedures were described previously [26]. We used Hanovia reactors for preparative experiments and a Moses merry-go-round [25] apparatus, immersed in a thermostatted water bath, for quantum yield measurements. We employed medium-pressure 200 W and 450 W Hg lamps. The 313 nm Hg line was isolated using a filter solution prepared by dissolving potassium chromate, 0.4 g, and sodium carbonate, 1.50 g, in 1.0 L. This solution was also used in the THF experiments [13]. The *trans* → *cis* photoisomerization of stilbene in pentane (*λ*_exc_ = 313 nm was used for actinometry), *ϕ*_tc_ = 0.52 [26]. Pyrex tubes, 13 mm o.d., fitted with standard taper joints and grease traps were loaded with 3.0 mL aliquots of solutions. These were degassed using 4–6 freeze-pump-thaw cycles to <10^−4^ Torr and flame-sealed at a constriction. All operations, including analyses, were performed under nearly complete darkness (red light). Ar, N_2_ and O_2_ outgassed solutions were used in some experiments.

## 4. Discussion

The results in Table 1 show that 63% of the HOCTL photoproducts involve reaction with ethanol. Furthermore, 47% of the photoproducts, HOE_1_, HOE_2_ and HOD_1_ have the same *s*-*trans* heteroannular diene moiety. This conjugated 1,3-diene accounts for the shift in the photoproduct UV *λ*_max_ from 260 nm in THF [13] to 244 nm in ethanol (Figure 1) [19]. With four ring residues and two exocyclic double bonds, the Woodward–Fieser rules [33,34] predict the *λ*_max_ of all three *s*-*trans*-1,3-dienes to be at 244 nm, very close to the *λ*_max_ in Figure 2 and Appendix A. Although HORP_1_ at 37% is the major photoproduct in degassed ethanol, it appears as a shoulder at ~260 nm in Figure 1 due to its significantly lower molar extinction coefficient [13].

### 4.1. Mechanism

Three mechanisms for the photoaddition of alcohols to olefins have been documented. One involves alkene photoionization followed by reaction of the radical cation with alcohol [35], the second involves photochemical formation of a high-energy ground state intermediate, such as a *trans*-cyclohexene [36,37], or a bicyclobutane [38,39] that reacts with alcohol, and the third involves trapping of an excited state twisted zwitterionic intermediate [40,41]. Dauben and Ritscher first proposed a zwitterionic excited diene state as the intermediate in the stereospecific photocyclization of *trans*-3-ethylidenecyclooctene to bicyclobutane [42]. Salem’s sudden polarization effect [43] close to orthogonal alkene geometries, confirmed recently by more advanced calculations on the stilbenes [44,45,46,47], provided theoretical support. Solvent effects on the lifetime of the tetraphenylethylene excited transient [48] as well as solvent [49,50,51,52,53] and substituent [54] effects on *cis*/*trans* photoisomerization, all support formation of twisted zwitterionic intermediates.

The photoreactions of CTL or HOCTL with ethanol are consistent with protonation of a zwitterionic excited state. The most relevant precedents for the addition and reduction photoproducts that we observed (Figure 2) are the toxisterols B and R that were obtained from the over-irradiation of 7-dehydrocholesterol (Pro) in ethanol or methanol [55,56]. The over-irradiation of ergosterol leads to similar results judging from the evolution of UV spectra obtained in the course of the irradiation [57]. The B molecules are ethers that are thought to arise by 1,6-addition of alcohol to the conjugated triene moieties of previtamin D, Pre, and/or tachysterol, Tachy, and R, an unconjugated diene, is the product of a competing photoreduction. Diallyl zwitterionic intermediates were envisioned that undergo protonation at C_9_ followed by alkoxide addition or hydride transfer [56]. Two twisted diallyl zwitterions should be accessible from *s*-*cis*,*s*-*trans*- and *s*-*cis*,*s*-*cis*-conformers of either Pre or Tachy (Figure 4). Protonation by methanol or ethanol at C_6_ of the zwitterions in Figure 4 would give the planar pentadienyl cations shown in Figure 7 and Figure 8 of [55] paired with methoxide or ethoxide anions.

The alkoxide counterion gives the observed ethers by coupling at C_10_ or the diene reduction products by hydride transfer at C_6_. Starting from different conformers of Pre and Tachy affords the desired pentadienyl cations [56] and avoids their proposed highly unlikely equilibration [57]. Havinga’s NEER principle now applies and there should be an excitation wavelength dependence on the composition of B and R toxisterols.

The alcohol photoreactions with CTL and HOCTL are simpler than those shown in Figure 4 because the structural rigidity of the conjugated triene chromophore eliminates *s*-*cis*-/*s*-*trans*-conformer equilibration and *cis*-/*trans*-isomerization from consideration. Three of the new photoproducts in ethanol, HOE_1_, HOE_2_ and HOD_1_ can form by protonation at C_5_ to yield an ion pair that collapses to the ether products or, via hydride transfer, to the diene, by reaction at C_6_ (Figure 5). Figure 5 shows the three resonance structures of the pentadienyl cation and the resonance hybrid. It also shows Pauling bond order calculations indicating that, if the ethoxide ion could freely move along the conjugated cation, the principle of least nuclear motion would favor reactions at C_8_ [58,59]. Because all three major photoproducts involve reaction of the ion pair at C_6_, it appears that the reaction is controlled by the proximity of the nascent ethoxide anion to the positive site [60]. Protonation at C_11_ gives a pentadienyl cation that could form the minor reduction photoproducts, HOPro and HOD_2_ (Figure 6). The proximity of the nascent ion pair favors hydride transfer at C_9_ to give HOPro, whereas the principle of least nuclear motion in the cation favors hydride transfer at C_7_ to give HOD_2_. We tend to discard the idea that two molecules of ethanol react with S_1_ of HOCTL in a concerted fashion, one to deliver a proton and the other to deliver ethoxide or hydride, because ethoxide should be a much better hydride donor than ethanol. Hydride transfer can give the endocyclic conjugated *s*-*cis*-1,3-diene, HOD_3_, shown on the right at the bottom of Figure 5 and Figure 6. Although not isolated, there is strong evidence that it is present. The vinyl protons of the known ^1^H NMR spectrum of this diene without the 25-OH group [23] appear as a pair of doublets at *δ* 5.817 and 5.377. They are almost coincident with the pair of doublets at *δ* 5.810 and 5.343 in the ^1^ H NMR spectrum of the product mixture (circled in Figure 5). Based on the area of the vinyl proton signals this product accounts for 8.2% of HOCTL loss.

### 4.2. Photochemical Kinetics

Except for the photoreactions with ethanol, the reaction sequences that account for the photochemistry of CTL or HOCTL in degassed or air-saturated ethanol are the same as for THF [13]. In both solvents, Equations (1)*–*(6) apply under degassed conditions, and Equations (7)*–*(10) are additional steps in the presence of O_2_. We do not show singlet oxygen, ^1^O_2_*, formation in Equation (7) because it was shown not to form in THF [13].
(1)C1TL→hnC1TL*
(2)C1TL*→kfC1TL+hn
(3)C1TL*→kp1P11
(4)C1TL*→kisC3TL*
(5)C1TL*→kdsC1TL
(6)C3TL*→kdtC1TL
(7)C1TL*+O32→koxsC3TL*+O32
(8)C3TL*+O32→koxtC1TL+O12*
(9)C1TL+O12*→kperCTLP
(10)O12*→kdoxO32

Under all conditions, P_e_ designates the sum of all ether and reduction photoproducts in ethanol (Equation (11)). Because their composition does not change as the reaction proceeds to completion, the sum of the pseudo unimolecular rate constants involved in their formation is designated *k*_e_.
(11)C1TL*+CH3CH2OH→kePe

### 4.3. Photophysics

The *ϕ*_f_/*τ*_f_ ratios in Table 3 give *k*_f_ = 1.7_2_ × 10^8^ s^−1^ at 20 °C (Equation (12)), where *τ*_s_ = (*k*_f_ + *k*_p_ + *k*_is_ + *k*_d_ + *k*_e_)^−1^. The significant deviation of this value from 1.4_5_ × 10^8^ s^−1^, the value we reported recently for HOCTL in THF at 20 °C [13] led us to re-examine the THF data in [13]. We found that while the lifetimes we had reported were indeed measured at 20 °C, the fluorescence quantum yields were measured at 22.8 °C. Table 3 shows that there is a significant increase in the *ϕ*_f_ of HOCTL on lowering the temperature from 25 to 20 °C. By interpolation, we expect a 14% increase in *ϕ*_f_ on lowering the temperature from 22.8 to 20.0 °C in ethanol. Applying this increase to the *ϕ*_f_ we reported for THF at 20 °C [13] under Ar-outgassed conditions gives *k*_f_ = 1.7_0_ × 10^8^ s^−1^, in excellent agreement with the values in ethanol in Table 3. It places our values in the middle of the wide *k*_f_ range that we calculated [13] from the data in Smutzer et al. in a variety of solvents [61] and well within experimental error of the value reported by Hyslop et al. in ethanol at 37 °C [62].
(12)$$\varphi_{f}^{}=k_{f}\tau_{f}^{}$$

The effect of O_2_ on fluorescence quantum yields and lifetimes is given by Equations (13) and (14), respectively, where *ϕ*_f_^0^ = *k*_f_*τ*_f_^0^ and *τ*_f_^0^ = (*k*_f_ + *k*_RP1_ + *k*_is_ + *k*_d_ + *k*_e_)*^−^*^1^ are the fluorescence quantum yield and lifetime in the absence of O_2_. Plots of the data in Table 3 according to Equations (13) and (14) are shown in Figure 7. The slope of the lifetime plot in Figure 7a gives $k_{ox}^{S}=4.9\times 10^{10}\;M^{-1}s^{-1}$ and the slope of the Stern–Volmer plot in Figure 7b gives $k_{ox}^{S}=5.1\times 10^{10}\;M^{-1}s^{-1}$. These values are the same within experimental error and indicate that oxygen quenching of ^1^HOCTL* in ethanol is diffusion controlled as in THF [13].
(13)$$\frac{\varphi_{f}^{0}}{\varphi_{f}^{}}=1+k_{ox}^{S}\tau_{f}^{0}[O_{2}]$$
(14)$$\frac{1}{\tau_{f}^{}}=\frac{1}{\tau_{f}^{0}}+k_{ox}^{S}[O_{2}]$$

The lowest excited singlet and triplet lifetimes of HOCTL in ethanol, 0.206 ns and 73.5 μs, respectively, are about 10% smaller than in THF [13]. The smaller ^1^HOCTL* lifetime in ethanol was expected due to the additional photoreactions with the solvent.

### 4.4. Photochemistry

Application of the steady-state approximation on all excited species in the above mechanism leads to the quantum yield expression in Equation (15) for rearrangement product formation, where k_p1_ is the rearrangement rate constant in Equation (3). Our quantum yield for HORP_1_ formation in degassed ethanol, 0.024, and *τ*_s_ = 0.20_6_ ns from Table 3 give *k*_P1_ = 1.1_7_ × 10^8^ s^−1^ for HOCTL, essentially identical to the value we obtained for the rearrangement rate constant in THF [13]. Quantum yields for the additional photoproducts in ethanol are given by Equation (16), where *k*_e_ is the pseudo-unimolecular rate constant for the trapping of ^1^HOCTL* by ethanol in Equation (11) and *f*_n_ are the fractions of the cations in Figure 5 and Figure 6 that give addition or reduction product P_n_. Assuming that ethanol trapping is exclusively by protonation and that cation formation is irreversible, *k*_e_ = 1.9_9_ × 10^8^ s^−1^. It is important to note that all identified ethanol photoproducts involve protonation at C_5_ or C_11_ the two terminal positions of the triene moiety with a strong preference for the C_5_ position. Those reactions give pentadienyl cations, confirming what JS was once told by Donald R. Arnold: “When a conjugated system gives a zwitterion, it does so by letting the positive charge occupy the biggest hole [63]”.
(15)$$\varphi_{P_{1}}^{}=k_{P_{1}}\tau_{f}$$
(16)$$\varphi_{P_{n}}^{}=f_{n}k_{e}\tau_{f}$$

When the irradiation is carried out in ethanol in the presence of O_2_, the formation of peroxide is subject to the same considerations that we described in detail for the reaction in THF [13]. Accordingly, the peroxide quantum yields in Table 2 were treated using Equation (17), where *τ*_Δ_ and *ϕ*_Δ_ are the lifetime and formation quantum yield of ^1^O_2_*, respectively.

The presence of 11% HOPro as a contaminant in the HOCTL used in this experiment causes a variation in *τ*_Δ_ as shown in Equation (18). HOPro competes for ^1^O_2_* but not for the light.
(17)$$\frac{1}{\phi_{HOCTLP}^{}}=(1+\frac{1}{k_{per}^{}\tau_{\Delta }^{}[\mathrm{HOCTL}]})\;\frac{1}{\phi \Delta }$$
(18)$$\tau_{\Delta }^{}=\frac{1}{k_{dox}^{}+k_{Pro}^{}[\mathrm{HOPro}]}$$

A summary of the many measurements of the lifetime, *τ*_dox_ = 1/*k*_dox_, of ^1^O_2_* in pure ethanol has been presented [64], and they are in reasonable agreement. We choose to use the largest of these, 15.3 ± 0.7 μs [65], because reaction of singlet oxygen with the sensitizer used to produce it may reduce its lifetime. The overall rate constant of the reactions of ^1^O_2_* with Pro or HOPro, *k*_Pro_, has not been measured in ethanol. Rate constants for the reactions of ergosterol [66] with ^1^O_2_* are 1.2_1_ × 10^7^ M^−1^s^−1^ and 2.1 × 10^7^ M^−1^s^−1^ in tert-butyl methyl ether [67] and benzene [68], respectively. As in the case of THF [13], we estimated *τ*_Δ_ values in ethanol by assuming that the reactivities of Pro and HOPro with ^1^O_2_* in ethanol are the same as the reactivity of ergosterol in tert-butyl methyl ether. Those *τ*_Δ_ values are given in the third column of Table 2 and the plot of the peroxide quantum yields according to Equation (17) is given in Figure 8.

The intercept/slope ratio of the plot of the inverse of the peroxide quantum yield against 1/(*τ*_Δ_[HOCTL]) in Figure 8 gives *k*_per_ = 2.9 × 10^7^ M^−1^s^−1^ in ethanol. The uncertainty in this value is large. It is larger than *k*_per_ = 1.0_7_ × 10^7^ M^−1^s^−1^, the value we had estimated in THF. Our THF value was based on the use of *τ*_Δ_ = 30 μs [69] for the lifetime of singlet oxygen in pure THF. With the use of *τ*_Δ_ = 20 μs, the value supported by more recent publications [70,71], we obtain *k*_per_ = 1.9_3_ × 10^7^ M^−1^s^−1^ in THF, in better agreement with our value in ethanol. The inverse of the intercept gives *ϕ*_Δ_ = *ϕ*_is_ = 0.16 in ethanol. With the use of *τ*_f_ = 0.20 ns from Table 3 we obtain *k*_is_ = 8._0_ × 10^8^ s^−1^ for HOCTL in ethanol, in close agreement with our value in THF [13]. We estimate the uncertainty in these values to be ±30%.

Our literature search for HOCTLP thermal rearrangement products was not fruitful. It appears that HOCTLP is quite stable and that the rearrangement in Figure 3 has not been reported previously. A plausible multistep mechanism for the reaction is given in Figure 7. Although some of the steps are combined in the Scheme, we do not wish to imply that they are necessarily concerted. The first intermediate shown in Figure 7 is a hydroperoxy-substituted carbocation. Although such species are common intermediates in reactions of peroxides, they are, counterintuitively, less stable than their hydroxy analogs [72,73]. Consequently, their subsequent transformation into (a) tertiary allylic cation and (b) oxacarbenium ion (i.e., the protonated ketone), is likely to be thermodynamically favorable. Note that the hydride shift shown in the second intermediate in Figure 7 would form an oxacarbenium ion. So the rearrangement proceeds from a less stable hydroperoxycarbocation to a more stable hydroxycarbocation—i.e., downhill thermodynamically. A recent general reference describing the stabilization of cations by oxygen is available [74].

The 7,6,6,5 A,B,C,D ring-size sequence in RP_2_ is unusual, as, to our surprise, we could find it in no reported natural product. Accordingly, we plan to seek optimum conditions for this reaction.

## Data Availability

Experimental data (electronic and emission spectra, NMR data) are available from the authors upon reasonable request.

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
