# Peer review of "Photochemistry and Photophysics of Cholesta-5,7,9(11)-trien-3β-ol in Ethanol"

_molecules, 2023, doi:10.3390/molecules28104086_

Round 1

Reviewer 2 Report

Review Report:

In the manuscript, Saltiel et al., report the photochemistry and photophysics of cholesta-5,7,9(11)-trien-3β-ol (CTL) in ethanol, which is a continuation of their previous work where they studied photochemistry in an aprotic environment. The results clearly explain the enhanced UV signal at 244 nm, resulting from the conjugated diene type photoproducts, which were absent in the aprotic THF solvent. Their results are also confirmed by other analytical methods described in the manuscript.

I find the work important, well-written and thorough. However, the presentation could be improved and there are some minor issues. I recommend publication of the present work after addressing those issues. I have used abbreviations, P-page number and L-line number. My comments are as follows:

Specific Comments:

1. P1-L37: The substituent should be written as OH and not as HO.

2. P2-Scheme2: It is hard to distinguish the photoproducts from one another. I recommend the authors to present the structures in a better way.

3. P4-Figure 1: There should be some paragraph spacing after the figure caption.

4. P5-L186: The authors mention that the UV spectra of HOD1 and HOE1 are similar. I find this is a bit hard to follow with different x-axis scaling and there is a positive signal for HOE1 after 265 nm, which is not there for the other species.

5. P5-Figure 2: I would like to recommend the authors to write the y-axis unit as au (absorbance unit). Again, there should be some paragraph spacing after the figure caption.

6. P7-L243: Maybe it is just me, I do not find what is actually RP2 I do not see a formal introduction of this species.

7. P7-L251: The authors introduce T as temperature without any prior introduction. They should clarify this type of abbreviation. This is also true for other abbreviations that they used, for instance, [t-St] and [c-St].

8. P8-Figure 5: The authors should work on this figure as it looks clearly stretched in one direction.

9. P9-L308: It should read as ‘Pulsed excitation” not as “Pulse excitation”.

10. P14-Figure 8: The presentation of the figure should be improved and consistent with previous figures (ticks’ direction).

General Comments:

1. The figure presentation should be improved.  

2. The authors should be consistent with the font and font size in the manuscript.

3. Sometimes, I find unwanted subscripts and spacings in the text. The authors should carefully check these things.
